# Computationally Optimized Hemagglutinin Proteins Adjuvanted with Infectimune^®^ Generate Broadly Protective Antibody Responses in Mice and Ferrets

**DOI:** 10.3390/vaccines12121364

**Published:** 2024-12-02

**Authors:** James D. Allen, Xiaojian Zhang, Jessica M. Medina, Matthew H. Thomas, Amanda Lynch, Ron Nelson, Julia Aguirre, Ted M. Ross

**Affiliations:** 1Center for Vaccines and Immunology, University of Georgia, Athens, GA 30605, USA; allenj25@ccf.org (J.D.A.);; 2Department of Infectious Diseases, University of Georgia, Athens, GA 30605, USA; 3Florida Research and Innovation Center, Cleveland Clinic, Port Saint Lucie, FL 34987, USA; medinaj7@ccf.org (J.M.M.); thomasm36@ccf.org (M.H.T.); lyncha9@ccf.org (A.L.); aguirrj3@ccf.org (J.A.); 4Department of Infection Biology, Lerner Research Institute, Cleveland Clinic, Cleveland, OH 44106, USA

**Keywords:** influenza, vaccine, infectimune, R-DOTAP, universal, hemagglutinin, H1N1, H3N2, broadly reactive, COBRA

## Abstract

**Background/Objectives:** Standard-of-care influenza vaccines contain antigens that are typically derived from components of wild type (WT) influenza viruses. Often, these antigens elicit strain-specific immune responses and are susceptible to mismatch in seasons where antigenic drift is prevalent. Thanks to advances in viral surveillance and sequencing, influenza vaccine antigens can now be optimized using computationally derived methodologies and algorithms to enhance their immunogenicity. **Methods:** Mice and ferrets that had been previously exposed to historical H1N1 and H3N2 influenza viruses were vaccinated intramuscularly with bivalent mixtures of H1 and H3 recombinant hemagglutinin (rHA) proteins, which were generated using a computationally optimized broadly reactive antigen (COBRA) design methodology. The vaccine antigens were mixed with a cationic lipid nanoparticle adjuvant, Infectimune^®^, which promotes both humoral and cellular immune responses. **Results:** Mice and ferrets vaccinated with Infectimune^®^ and COBRA rHAs elicited protective antibody titers against panels of H1N1 and H3N2 influenza viruses isolated over the past 10 years. These animals also had antibodies that neutralized numerous modern H1N1 and H3N2 influenza viruses in vitro. When challenged with the A/Victoria/2570/2019 H1N1 influenza virus, the COBRA rHA vaccinated animals had minimal weight loss, and no detectable virus was present in their respiratory tissues on day 3 post-infection. **Conclusions:** These results demonstrate that COBRA rHA vaccines formulated with Infectimune^®^ elicit protective antibody responses against influenza strains, which were isolated across periods of time when standard-of-care vaccines were frequently reformulated, thus reducing the need to update vaccines on a nearly annual basis.

## 1. Introduction

Diseases caused by seasonal influenza viruses are a significant public health concern that causes both an economic impact and a societal burden, which stem from health care costs and missed work or school [1]. Vaccination is considered the most effective way to combat influenza, and the US Centers for Disease Control and Prevention (CDC) estimates that vaccination prevented ~105,000 hospitalizations and 6300 deaths during the 2019–2020 influenza season [2,3]. However, seasonal vaccines vary in effectiveness from season to season, depending on how well the selected antigens match what is circulating in the human population [4,5]. The CDC estimated that during the 2023–2024 influenza season, vaccine efficacy in adults over 18 years old ranged from ~33 to 49% [6]. Thus, there is a need for more effective vaccines that can protect against more viruses within and across each season [7,8,9].

Antigen selection plays a large role in vaccine effectiveness from season to season [10]. Typically, the World Health Organization (WHO) selects vaccine antigens on predictions of predominant clades based on current viral surveillance, but these antigens are selected ~6 months in advance of the upcoming influenza season to ensure enough time to manufacture the necessary number of doses [9,11,12]. What complicates matters is that influenza is an antigenically variable virus that can change its immunogenic glycoproteins to escape host immunity over time due to a process known as antigenic drift, which can also impact seasonal vaccine efficacy [13,14]. However, due to advances in viral surveillance and sequencing, vaccine antigens can now be quickly designed using in silico-derived methodologies to enhance their immunogenicity against antigenically drifted strains [15,16,17,18]. Computationally optimized broadly reactive antigens (COBRAs) for influenza hemagglutinin (HA) elicit protective immune responses across panels of antigenically distinct H1N1, H3N2, and H5N1 influenza viruses in mice, ferrets, and nonhuman primates [19,20,21,22]. The COBRA methodology is a layered consensus-building approach for designing vaccine antigens that utilize hemagglutinin (HA) sequence information from recently circulating viruses to generate HAs that contain immunogenic epitopes from multiple viruses [23]. Immunizing people with broadly reactive antigens that contain a diverse set of immunogenic epitopes could enhance vaccine efficacy by recalling a more heterogeneous population of memory B and T cells than traditional stain-specific antigens [24].

Vaccine effectiveness can also be improved through the use of adjuvants, which work to enrich antigen uptake and presentation by immune cells [25]. Adjuvants are currently used in vaccines tailored for older adults ≥65 years old to help mitigate the effects of immunosenescence as these individuals are more susceptible to influenza virus-associated illness, hospitalization, and death, compared to younger adults [26]. Nevertheless, adjuvants could be included in vaccine formulations for all age groups to increase vaccine immunogenicity. MF59, an oil-in-water emulsion adjuvant, increases the magnitude and breadth of immune responses in vaccinated individuals and is used in commercial vaccines, such as Fluad^®^, which is currently marketed toward the elderly population [2,27]. Adjuvants such as Alum and MF59 are approved for use in human vaccines in the U.S., but these adjuvants typically induce T cell helper type 2 (Th2)-biased humoral immune responses [27,28]. Thus, adjuvants that elicit more balanced Th1 and Th2 immune responses are currently under investigation. One such adjuvant, Infectimune^®^, is an enantiospecific cationic lipid nanoparticle that elicits potent humoral and cellular immune responses and has also demonstrated safety in Phase I and Phase II clinical trials (NCT02065973, NCT04260126, NCT04580771, NCT05232851) [29,30,31].

In this study, COBRA recombinant HA (rHA) proteins specific for H1 and H3 were combined with Infectimune^®^ to determine if the adjuvant could increase the antibody responses elicited by the rHA vaccines, compared to unadjuvanted formulations, in a population of mice that had been previously exposed to H1N1 and H3N2 influenza. In addition to enhancing antigen uptake and presentation, adjuvants can also reduce the amount of vaccine antigen necessary to elicit protective immune responses [32]. Therefore, mice were vaccinated with different doses of COBRA rHA to investigate the dose-sparing effects of the adjuvant. Lastly, COBRA rHA vaccine and adjuvant formulations were compared head-to-head with wild type rHA vaccines in the ferrets that had previously been infected with historical H1N1 and H3N2 isolates. Collected sera were then assessed for hemagglutination-inhibition (HAI) and neutralization activity against panels of modern influenza vaccine strains. Overall, the results herein highlight the potential benefits of combining computationally optimized vaccine antigens with novel immunostimulatory adjuvants to broaden immune responses, reduce vaccine dose, and increase vaccine efficacy.

## 2. Materials and Methods

### 2.1. Recombinant HA Proteins

The COBRA H1 Y2 HA amino acid sequence was synthesized from wild type H1 HA sequences that circulated from 2014 to 2016 [33]. The COBRA NG2 HA sequence was synthesized from wild type H3 HA sequences that circulated from 2016 to 2018 [34]. Amino acid sequences for the H1 A/Michigan/45/2015 (Mich/15) HA (EPI_ISL_205499) and the H3 A/Singapore/IFNIMH-16-0019/2016 (Sing/86) HA (EPI_ISL_285898) were downloaded from the Global Initiative on Sharing Avian Influenza Data (GISAID) EpiFlu online database. Each HA amino acid sequence was truncated to replace the transmembrane domain with a solubilization domain, a T4 fold-on domain, an AviTag, and a 6× histidine tag. Soluble recombinant HA (rHA) proteins were then generated by individually cloning each truncated HA sequence into the pcDNA3.1^+^ plasmid (Azenta, South Plainfield, NJ, USA). Each plasmid was individually transfected into HEK293T suspension cell cultures as previously described [35]. The soluble rHA proteins were then purified via his-tag purification using an AKTA pure chromatography system (Cytiva, Marlborough, MA, USA). Following purification, the concentration of each soluble rHA protein was determined using a conventional bicinchoninic acid assay (BCA) kit (Thermo Fisher, Waltham, MA, USA). The purified rHA proteins were used as vaccine antigens to coat ELISA plates for serological assays. The recombinant HA vaccine antigens were formulated with Infectimune^®^ adjuvant (PDS Biotechnology, Princeton, NJ, USA) unless otherwise noted. Infectimune^®^ is an R-enantiomer of the cationic lipid 1,2-dioleoyl-3-trimethylammonium-propane (R-DOTAP) [29].

### 2.2. Viruses

Influenza viruses were obtained from either the U.S. Centers for Disease Control (CDC), the Biodefense and Emerging Infections Resources Repository (BEI Resources), the Influenza Reagents Resource (IRR), or provided by Virapur (San Diego, CA, USA). The viruses were passaged in embryonated chicken eggs following World Health Organization (WHO) recommended protocols [36]. Upon receipt, virus stocks were quantified via hemagglutination assay, made into single-use 1 mL aliquots, and stored at −80 °C for future use. The H1N1 viruses used in this study were: A/California/07/2009 (Cal/09) (clade pdm09), A/Michigan/45/2015 (Mich/15) (clade 6B.1), A/Brisbane/02/2018 (Bris/18) (clade 6B.1a.1), A/Guangdong-Maonan/SWL/1536/2019 (Guang/19) (clade 6B.1A.5a.2), and A/Victoria/2570/2019 (Vic/19) (clade 6B.1A.5a.2). The H3N2 viruses used in this study were: A/Switzerland/9715293/2013 (Switz/13) (clade 3c.3a), A/Hong Kong/4801/2014 (HK/14) (clade 3c.2a), A/Singapore/IFNIMH-16-0019/2016 (Sing/16) (clade 3c.2a.1), A/Switzerland/8060/2017 (Switz/17) (clade 3c.2a2), A/Kansas/14/2017 (Kan/17) (clade 3c.3a), A/South Australia/34/2019 (SA/19) (clade 3c.2a1b.2), A/Hong Kong/2671/2019 (HK/19) (clade 3c.2a1b.1b), and A/Tasmania/503/2020 (Tas/20) (clade 3c.2a1b.2a.1). All viruses used in this study were historical WHO recommended vaccine strains.

### 2.3. Vaccination and Viral Infection of Mice

Influenza naïve female BALB/c mice, *Mus musculus*, (6–8 weeks old, *n* = 48) were obtained from The Jackson Laboratory (Bar Harbor, ME, USA). The mice were housed in microisolator caging, provided access to water and food ad libitum throughout the duration of the study, and cared for following the USDA guidelines for laboratory animals. The procedures used in this study were approved by the University of Georgia Institutional Animal Care and Use Committee (no. A2021-06-020-Y3-A1). Prior to beginning the study, the mice were bled to confirm that they did not have any pre-existing antibodies specific to influenza. After this was confirmed, the mice were randomly divided into 6 groups of 8 animals on day 0 and infected intranasally with 50 μL of a mixture of 2.5 × 10^5^ PFU/mL of A/Singapore/6/1986 (Sing/86) H1N1 virus and 2.5 × 10^5^ PFU/mL of A/Panama/2007/1999 (Pan/99) H3N2 virus diluted in sterile phosphate-buffered saline (PBS) (Thermo Fisher, Waltham, MA, USA). The mice were weighed and monitored daily for 14 days following infection to track the progression of the disease. On day 14, blood was collected from each mouse and was used to confirm that the animals had all achieved an HAI titer of 1:40 or higher against both the Sing/86 and Pan/99 viruses. The mice were then allowed to rest for 30 days and then were vaccinated intramuscularly with different doses of COBRA rHA proteins. Group 1 was vaccinated with 3 μg of Y2 and 3 μg of NG2 resuspended in sterile sucrose in a total volume of 50 μL. Group 2 was vaccinated with 25 μL of sterile sucrose mixed 50:50 *v*/*v* with Infectimune^®^ adjuvant (6 mg/mL) for a total volume of 50 μL. Group 3 was vaccinated with 3 μg of Y2 and 3 μg of NG2 resuspended in sterile sucrose mixed 50:50 *v*/*v* with Infectimune^®^ adjuvant (6 mg/mL) in a total volume of 50 μL. Group 4 was vaccinated with 0.6 μg of Y2 and 0.6 μg of NG2 resuspended in sterile sucrose mixed 50:50 *v*/*v* with Infectimune^®^ (6 mg/mL) in a total volume of 50 μL. Group 5 was vaccinated with 0.12 μg of Y2 and 0.12 μg of NG2 resuspended in sterile sucrose mixed 50:50 *v*/*v* with Infectimune^®^ (6 mg/mL) in a total volume of 50 μL. Group 6 was vaccinated with 0.024 μg of Y2 and 0.024 μg of NG2 resuspended in sterile sucrose mixed 50:50 *v*/*v* with Infectimune^®^ (6 mg/mL) in a total volume of 50 μL. The mice were vaccinated in a prime-boost regimen on day 30 and day 58, and 14 days after each vaccination, blood was collected from every mouse. On day 86, the mice were challenged intranasally with A/Victoria/2570/2019 H1N1 influenza virus by allowing anesthetized mice to inhale 50 μL of the virus at a dose of 2.7 × 10^6^ PFU (plaque forming units)/50 μL. The mice were then monitored for 14 consecutive days for weight loss and clinical symptoms. On day 89, lungs were collected from 3 mice in each group to determine the amount of replicating virus present in the respiratory tissue 3 days post-infection. On day 100, the mice were humanely euthanized.

### 2.4. Vaccination and Viral Infection of Ferrets

Influenza naïve female Fitch Ferrets, *Mustela furo*, (6–8 months old, *n* = 18) were obtained from Triple F Farms (Gillet, PA, USA) and divided into 3 groups of 6 animals. During most of the study, the ferrets were pair-housed in open-air caging (Allentown LLC, Allentown, NJ, USA). When the animals were exposed to the influenza virus, they were pair-housed and moved into Type 1800 individually ventilated caging (Allentown LLC, Allentown, NJ, USA) for 14 days. The ferrets were provided food and water ad libitum for the duration of the study and were cared for following the USDA guidelines for laboratory animals. The procedures used in this study were reviewed and approved by the Cleveland Clinic Institutional Animal Care and Use Committee (Protocol no. 2948). Prior to beginning the study, the ferrets were bled to confirm that they did not have any pre-existing antibodies specific to influenza. After this was confirmed, on day 0 the animals were randomly divided into 3 groups (n = 6/group) and infected intranasally with 1 mL of sterile PBS containing 5 × 10^5^ PFU/mL of A/Singapore/6/1986 (Sing/86) H1N1 virus and 5 × 10^5^ PFU/mL of A/Panama/2007/1999 (Pan/99) H3N2 virus. The ferrets were weighed and monitored daily for 14 days following infection to monitor the progression of the disease. On day 14, blood was collected from the ferrets and was used to confirm that the animals had all achieved an HAI titer of 1:40 or higher against both the Sing/86 and Pan/99 viruses. The ferrets were allowed to rest for 60 days post-infection and were then vaccinated intramuscularly in the hind leg on day 60 with 250 μL of sterile sucrose containing 15 μg of Y2 and 15 μg of NG2 or 15 μg of Mich/15 and 15 μg of Sing/16 mixed 50:50 *v*/*v* with Infectimune^®^ adjuvant (6 mg/mL). The ferrets were boosted with homologous vaccine formulations on day 88, and blood was collected 14 days following each vaccination (day 74 and day 102) for use in serological assays. On day 116, the ferrets were challenged intranasally with 1 mL of sterile PBS containing 1 × 10^6^ PFU/mL of A/Victoria/2570/2019 H1N1 virus. The ferrets were then monitored for 14 consecutive days for weight loss and clinical symptoms. On day 119, nasal wash samples were collected from all the animals using 3 mL of sterile PBS to determine the amount of replicating virus present in the respiratory tract 3 days post-infection. On day 130, the ferrets were humanely euthanized. One ferret from the Y2 and NG2 vaccinated group and one animal from the mock vaccinated group were removed from the study due to non-study-related veterinary reasons, resulting in a group size of *n* = 5 for these two groups.

### 2.5. Hemagglutination Inhibition Assay (HAI)

The HAI assay was used to quantify the presence of antibodies that inhibit the binding of the viral HA glycoprotein to sialic acids present on the surface of red blood cells (RBCs). In this study, HAI assays performed with H1N1 viruses were assessed with turkey RBCs (Lampire Biological Laboratories, Pipersville, PA, USA), and assays performed with H3N2 viruses were run with guinea pig RBCs (Lampire Biological Laboratories, Pipersville, PA, USA). Prior to performing the assay, serum samples were treated with a receptor-destroying enzyme (RDE) (Denka, Seiken, Co., Tokyo, Japan). First, three volumes, 300 μL, of RDE reconstituted in PBS was mixed with one volume,100 μL, of sera. This combination was then incubated at 37 °C for 16 h. The samples were then placed in a 56 °C water bath for half an hour. After heat treatment, six volumes, 600 μL, of PBS were added to each serum sample to create a 1:10 serum dilution. Following RDE treatment, 50 μL of each 1:10 diluted sera sample was individually added to different wells in the first column of a 96-well V-bottom microtiter plate (Thermo Fisher, Waltham, MA, USA), and 25 μL of PBS was added to the remaining wells of the plate. The sera samples were diluted two-fold across the plate by transferring 25 μL from one well to the next, discarding the final 25 μL. An equal volume, 25 μL, of influenza virus, previously tittered to 8 hemagglutination units/50 μL in PBS, was added to each well. For H1N1 viruses, the plates were incubated for 20 min, and for H3N2 viruses, the virus solutions were supplemented with 20 nM Oseltamivir carboxylate (Aobious, Gloucester, MA, USA), and the plates were allowed to incubate for 30 min. Next, 50 μL of washed turkey RBCs diluted to 0.8% in PBS were added to the plates containing H1N1 viruses and allowed to incubate for 30 min. Similarly, 50 μL of guinea pig RBCs were diluted to 0.8%, added to plates containing H3N2 viruses, and incubated for 1 h. The plates were then titled to observe the presence or absence of hemagglutination, and the HAI titer of the serum was determined by taking the reciprocal of the dilution in the last well that contained nonagglutinated RBCs. In this study, a “protective” antibody titer is described as possessing an HAI titer of ≥1:40, as per the WHO and European Committee for Medicinal Products guidelines for evaluating influenza vaccines [36].

### 2.6. Enzyme-Linked Immunosorbent Assay (ELISA)

Immulon 4HBX 96-well microtiter plates (Thermo Fisher, Waltham, MA, USA) were coated with 1 μg/mL of Y2 or NG2 rHA by mixing the protein with carbonate coating buffer pH 9.4 (2.65 g Na_2_CO_3_, 2.1 g NaHCO_3_, 450 mL diH_2_O) and adding 100 μL of the solution to each well. The plates were then incubated overnight at 4 °C in a humidified chamber. Following incubation, the coating solution was removed, and there was 200 μL of blocking buffer (PBS, 4% FBS (fetal bovine serum), 0.05% Tween-20 (Thermo Fisher, Waltham, MA, USA)) was added to each well of the plates. The blocking buffer was allowed to incubate in the plates at 37 °C for 90 min. In a separate 96-well V-bottom microtiter plate (Thermo Fisher, Waltham, MA, USA) 120 μL of blocking buffer was added to every well to set up a dilution plate. An extra 24 μL of blocking buffer and 36 μL of vaccinated animal serum pooled from each vaccine group diluted 1:100 was added to row A of the plate. Next, 60 μL from row A was serially transferred down the plate, creating a series of 2-fold dilutions, and the last 60 μL from row G was discarded. Similarly, an extra 24 μL of blocking buffer and 36 μL of murine reference sera was added to well H1, and 60 μL was serially transferred across the row from H1 to H9. Wells H10–H12 were used as negative controls with no sera and only contained 120 μL of blocking buffer. Following incubation, the blocking buffer was decanted from the protein-coated plates, and 100 μL from each well of the dilution plate was transferred to the protein-coated plate. The plates were then moved to a humidified chamber and incubated at 37 °C for 90 min. Once this time had passed, the plates were decanted and washed five times with wash buffer (PBS + 0.05% Tween-20). After washing, the plates were decanted, and 100 μL of secondary antibody diluted 1:4000 in blocking buffer was added to each well. Three different secondary antibodies were used in this study and were added to separate assay plates: Goat anti-mouse IgG HRP (1 mg/mL) (Cat. no. 1030-05, Southern Biotech, Birmingham, AL, USA), Goat anti-mouse IgG1 HRP (1 mg/mL) (Cat. no. 1070-05, Southern Biotech, Birmingham, AL, USA), Goat anti-mouse IgG2a HRP (1 mg/mL) (Cat. no. 1083-05, Southern Biotech, Birmingham, AL, USA). After the addition of the secondary antibody, the plates were moved to a humidified chamber and incubated at 37 °C for 90 min. Following incubation, the plates were decanted and 100 μL of substrate (1 mg/mL ABTS diammonium salt (Sigma, Burlington, MA, USA), McIlvaine solution pH 5 (7.31 g Na_2_HPO_4_ + 4.66 g C_6_H_8_O_7_ + 500 mL diH_2_O), 3% H_2_O_2_) was added to each well and allowed to develop for 12–15 min at 37 °C in a humidified chamber. After colorimetric development, 50 μL of a 1% SDS (sodium dodecyl sulfate) (Thermo Fisher, Waltham, MA, USA) solution was added to each well to stop the reaction. The plates were then read at 414 nm using a BioTek Epoch 2 plate reader (Agilent, Santa Clara, CA, USA) equipped with Gen6 software version 1.03. The endpoint dilution was determined by taking the reciprocal dilution of the most dilute well that contained more than twice the mean signal of the background (negative control) wells.

### 2.7. Microneutralization Assay

Prior to the assay, pooled sera samples from each vaccinated group of ferret samples were prepared and heat treated at 56 °C for 45 min. On the day of the assay, each well of a 96-well flat bottom microtiter plate (Thermo Fisher, Waltham, MA, USA) was filled with 50 μL of virus diluent (Dulbecco’s Modified Eagle Medium (DMEM), 1% Penicillin/Streptomycin, 13.5% Bovine Serum Albumin (BSA), 2.5% 1 M HEPES buffer, 2 μg/mL TPCK trypsin (Thermo Fisher, Waltham, MA, USA). An additional 40 μL of virus diluent was added to the first column of the microtiter plate, followed by the addition of 10 μL of pooled heat-treated serum. A series of two-fold dilutions was then established by transferring 50 μL of fluid from the first row across the plate and discarding the final 50 μL at column 10. Next, 50 μL of virus, which has been previously tittered to 100× the 50% tissue culture infectious dose (100× TCID_50_), was added to each well, excluding column 12. An extra 50 μL of virus diluent was added to column 12 to establish negative control wells. The plates were then incubated at 37 °C for 1 h. Following incubation, 100 μL of Madin-Darby canine kidney SIAT-1 cells (MDCK-SIAT) at a concentration of 1.5 × 10^5^–2.0 × 10^5^ cells/mL in virus diluent were added to every well of the plate. The plates were then incubated at 37 °C + 5% CO_2_ for 18–20 h. The next day, the media was removed from the plates, and each well was washed with 200 μL of PBS. The PBS was then removed, and the wells were fixed with ice-cold fixative solution (80% Acetone (Thermo Fisher, Waltham, MA, USA) diluted in PBS and incubated at room temperature for 10 min. The fixative solution was then decanted, and the plates were allowed to air dry for 1 h. The plates were then washed three times with 200 μL wash buffer (PBS + 0.03% Tween-20 (Thermo Fisher, Waltham, MA, USA)). The plates were then decanted, and 100 μL of rabbit anti-influenza-A-NP polyclonal antibody (1 mg/mL, Cat. no. PA5-81661, Thermo Fisher, Waltham, MA, USA) diluted 1:3000 in blocking buffer (PBS + 0.03% Tween-20 + 14.5% BSA) was added to each well. The plates were then moved to a humidified chamber and allowed to incubate at room temperature for 1 h. The plates were then washed three times with 200 μL wash buffer. After washing, the plates were decanted, and 100 μL of goat-anti-rabbit IgG (H+L) HRP (1 mg/mL, Cat. no. 31460, Thermo Fisher, Waltham, MA, USA) diluted 1:5000 in blocking buffer was added to each well. After the addition of the secondary antibody, the plates were moved to a humidified chamber and incubated at room temperature for 1 h. Following incubation, the plates were washed five times with 200 μL wash buffer. The wash buffer was then decanted, and 100 μL of the substrate (phosphate citrate buffer (Sigma, Saint Louis, MO, USA), o-phenylenediamine dihydrochloride (OPD) tablets (Sigma, Saint Louis, MO, USA), 0.05% 30% H_2_O_2_) was added to each well and allowed to develop for 3–5 min at room temperature. Once a color change started to occur in the negative control wells, column 12, 100 μL of stop solution (2N H_2_SO_4_ (Thermo Fisher, Waltham, MA, USA)) was added to each well. The plates were then read at 492 nm using a BioTek Epoch 2 plate reader (Agilent, Santa Clara, CA, USA) equipped with Gen6 software version 1.03. The 50% neutralization titer was determined by averaging the optical density (OD) values of the virus control wells in column 11 minus the background signal, which is the average of column 12, and using that value as the 100% infection titer. The values in the experimental wells were then compared to the 100% infection OD value, and a 50% inhibition titer was calculated for each experimental group using a linear interpolation of the Log2 serum titers above and below the 50% neutralization OD value. All plates were run in duplicate, and the neutralization titers were averaged between the plates.

### 2.8. Influenza Virus Plaque Assay

Influenza virus plaque assays were employed to quantify the replicating virus present in both the mouse and ferret respiratory tracts at 3 days post-infection. In brief, 1 × 10^6^ Madin-Darby Canine Kidney (MDCK) epithelial cells (Sigma, Saint Louis, MO, USA) were added to each well of a 6-well culture plate (Thermo Fisher, Waltham, MA, USA) and were incubated overnight at 37 °C. During the infections, lung samples were collected from 3 mice in each group on day 3 post-infection and stored at −80 °C. On the day of the assay, lung samples were thawed and homogenized in 1 mL of DMEM containing 1% Penicillin/Streptomycin (DMEM + P/S) (Thermo Fisher, Waltham, MA, USA). The homogenates were then pelleted by centrifugation at 2000 rpm for 5 min, and the supernatants were harvested. Likewise, nasal wash samples were collected from each ferret on day 3 post-infection, stored at −80 °C for future use, and thawed on the day the plaque assay was performed. Once thawed, the samples were serially diluted 10-fold in DMEM + P/S. When the MDCK cells reached 90–95% confluency in the 6-well plates, they were rinsed 2x with DMEM + P/S. Following the wash steps, the media was removed, and 100 μL of each dilution of mouse lung homogenate or ferret nasal wash was added to the individual wells of the plates in duplicate. The plates were allowed to incubate at room temperature for 1 h. During this time, the plates were gently shaken every 15 min so that the wells did not dry. Once the hour-long incubation was complete, the media was discarded, and the wells were washed 2× with DMEM + P/S. The media was decanted, and 2 mL of a mixture (50:50 *v*/*v*) of 2×MEM and 1.6% agarose supplemented with 1 μg/mL of TPCK trypsin (Thermo Fisher, Waltham, MA, USA) was added to each well. Once the solution solidified and cooled to room temperature, the plates were moved to an incubator set to 37 °C + 5% CO_2_ for approximately 72 h. After 72 h, the agarose gels were removed, and the cells were fixed by adding 1 mL of 10% buffered formalin (Thermo Fisher, Waltham, MA, USA) to each well. After 10 min, the formalin was decanted, and all the wells were stained with 0.5 mL of 1% crystal violet solution (Thermo Fisher, Waltham, MA, USA) for 10 min. The crystal violet was then decanted from each well; the plates were rinsed 5× with cold tap water and allowed to air dry. Once completely dry, the viral plaques were counted and recorded as the number of plaques present in the reciprocal of each dilution. Viral titers were reported as PFU/mL of sample for each mouse or ferret sample.

## 3. Statistical Analysis

Statistical differences between sample groups were determined using an ordinary nonparametric Kruskal–Wallis one-way analysis of variance (ANOVA) test comparing the mean rank of each column with the mean rank of every other column in GraphPad Prism software version 10.2.1 (GraphPad, San Diego, CA, USA). In this study, a *p* value < 0.05 was defined as statistically significant (* = *p* < 0.05, ** = *p* < 0.01, *** = *p* < 0.001, **** = *p* < 0.0001).

## 4. Results

### 4.1. Infectimune^®^ Adjuvant Elicited Enhanced HA Specific Antibody Responses Compared to Unadjuvanted rHA Vaccines

A dose-sparing study was performed to determine the ability of Infectimune^®^ to improve antibody titers generated by COBRA rHA vaccines. Mice that had been previously exposed to historical H1N1 and H3N2 influenza viruses were vaccinated twice with different quantities of the rHA vaccine that was administered with or without adjuvant. Two weeks after the second vaccination, blood was collected from the mice, and the resulting sera samples were tested for HAI activity against panels of historical H1N1 and H3N2 vaccine strains (Figure 1 and Figure 2). Mice vaccinated with either the 3 μg or 0.6 μg doses of the Y2 and NG2 COBRA rHA formulated with Infectimune^®^ adjuvant had mean protective (≥1:40) HAI titers against all the historical H1N1 vaccine strains from 2009 to 2019. These titers were also significantly (*p* < 0.05) higher than mice vaccinated with 3 μg of unadjuvanted rHA, the mock vaccinated mice, and the mice vaccinated with 0.024 μg of rHA adjuvanted with Infectimune^®^ (Figure 1A–E). Mice vaccinated with 0.12 μg of Y2 and NG2 rHA had mean protective HAI titers against 3 of the 5 H1N1 viruses in the panel. The HAI titers in this set of animals were also ~4-fold lower than the titers observed in the 3 μg and 0.6 μg groups that were adjuvanted with Infectimune^®^. Animals in the 0.12 μg group had similar HAI titers to mice vaccinated with 3 μg of rHA without adjuvant (Figure 1A–E). Mice vaccinated with 0.024 μg of Y2 and NG2 rHA and the mock vaccinated animals did not have protective HAI titers against any of the H1N1 viruses (Figure 1A–E).

Against the H3N2 virus panel, animals immunized with either 3 μg or 0.6 μg of the Y2 and NG2 COBRA rHA formulated with Infectimune^®^ adjuvant had mean protective HAI titers against 5 of the 8 strains. The HAI titers against these 5 viruses were, on average, ~2–4-times higher than the mice vaccinated with 3 μg of unadjuvanted rHA and the mice vaccinated with 0.12 μg of rHA formulated with Infectimune^®^. Additionally, mice in these two groups had significantly higher HAI titers than the pre-immune mock-vaccinated animals against these 5 viruses (Figure 2A–H). Similar HAI titers were observed across the H3N2 panel in the groups vaccinated with 0.12 μg of rHA formulated with Infectimune^®^ and 3 μg of rHA that was not adjuvanted. The mice in these two groups had mean protective titers against 2 of the 8 strains in the panel, Switz/17 and SA/19, and the HAI titers between these two groups were often within ~2-fold of one another (Figure 2A–H). The vaccines containing 0.024 μg of rHA formulated with Infectimune^®^, on average, did not induce sero-protective HAI titers against any of the viruses in the H3N2 panel. In this group, the highest HAI titers were elicited against the Switz/17 and Tas/20 viruses (Figure 2A–H). None of the pre-immune mock vaccinated animals had protective HAI titers against any of the viruses in the H3N2 panel (Figure 2A–H).

The serum samples collected after the second vaccination were then pooled for each group and tested via ELISA to determine the total IgG and antibody IgG isotype subclasses elicited by each of the COBRA rHA vaccines (Figure 3). Mice vaccinated with 3 μg and 0.6 μg doses of Y2 and NG2 rHA formulated with Infectimune^®^ had anti-Y2 HA-specific total IgG titers that were ~2–4-fold higher than the unadjuvanted group and the mice immunized with 0.12 μg of adjuvanted rHA (Figure 3A). The anti-Y2 IgG1 antibody titers were similar across the adjuvanted 3 μg, 0.6 μg, 0.12 μg, and unadjuvanted groups and were ~6–8-fold higher than the adjuvanted 0.024 μg group (Figure 3B). The IgG2a antibody titers against Y2 were ~6–8 times higher in the 3 μg and 0.6 μg adjuvanted groups than the unadjuvanted and 0.12 μg groups (Figure 3C). Mice vaccinated with adjuvanted doses of 3 μg, 0.6 μg, and 0.12 μg of rHA had similar NG2 specific total IgG titers to the unadjuvanted rHA vaccinated mice, and all these groups had IgG titers that were ~4-fold higher than the mice vaccinated with the adjuvanted 0.024 μg rHA dose (Figure 3D). The anti-NG2 IgG1 antibody titers were similar between the unadjuvanted and the adjuvanted 3 μg and 0.6 μg rHA vaccinated groups. The IgG1 titers in the adjuvanted 0.12 μg and 0.024 μg groups were ~2–4 fold lower than the unadjuvanted mice (Figure 3E). The adjuvanted 3 μg and 0.6 μg rHA vaccinated mice had NG2 specific IgG2a antibody titers that were ~2-fold higher than those in the unadjuvanted mice and the mice vaccinated with either 0.12 μg or 0.024 μg of rHA formulated with Infectimune^®^ (Figure 3F). Mice vaccinated with 3 μg of unadjuvanted rHA had ~6 times higher H1 specific anti-Y2 IgG1 antibody titers than anti-Y2 IgG2a titers. However, animals vaccinated with 3 μg of rHA adjuvanted with Infectimune^®^ had a ~1:1 ratio of anti-Y2 IgG1 and IgG2a antibodies. The mice vaccinated with 0.6 μg, 0.12 μg, and 0.024 μg of adjuvanted rHA also had a ~2–4-fold reduction in their H1 specific IgG1:IgG2a ratio compared to the unadjuvanted rHA vaccinated group (Figure 3G). Additionally, mice vaccinated with the unadjuvanted 3 μg dose of rHA had a ~2:1 ratio of H3-specific anti-NG2 IgG1:IgG2a antibodies. Animals vaccinated with COBRA rHA formulated with Infectimune^®^ had IgG1:IgG2a ratios that were ~1:1 (Figure 3H). Taken together, these data suggest that formulating COBRA rHA vaccines with Infectimune^®^ allows for the induction of broadly protective HAI titers against H1N1 and H3N2 viruses at a 5× lower dose than COBRA rHA alone and that these adjuvanted vaccines elicit a more balanced Th1/Th2 antibody response than unadjuvanted rHA.

### 4.2. COBRA rHA Vaccines Adjuvanted with Infectimune^®^ Protected Mice from H1N1 Infection

The mice were infected on day 86 post-vaccination with 2.7 × 10^6^ PFU of A/Victoria/2570/2019 H1N1 virus and were tracked for 14 days for weight loss and clinical symptoms of infection (Figure 4). Body weight loss across the groups of mice peaked on day 3 post-infection, and most of the animals returned to their starting body weights by day 7 post-infection. The most weight loss was observed in the pre-immune mock vaccinated mice and the mice vaccinated with 0.024 μg of rHA, formulated with Infectimune^®^, which lost ~10–18% body weight. The mice vaccinated with unadjuvanted rHA and 0.12 μg of adjuvanted rHA lost ~7–16% body weight, and mice vaccinated with 3 μg and 0.6 μg of adjuvanted rHA lost ~4–12% body weight. Mice vaccinated with 3 μg of adjuvanted rHA had significantly less weight loss on day 2 post-infection compared to the mice in every other group (*p* < 0.01), apart from the mice vaccinated with 0.6 μg of adjuvanted rHA. Mice in the 3 μg adjuvanted group also had significantly less weight loss on day 3 post-infection compared to mice vaccinated 0.024 μg of adjuvanted rHA and the pre-immune mock vaccinated mice (*p* < 0.05) (Figure 4A). During infection, clinical scores associated with lethargy were observed in the pre-immune mock and nonadjuvanted mice, as well as the adjuvanted 0.12 μg and 0.24 μg rHA vaccinated mice. Most clinical scores were recorded on day 3 post-infection and had resolved by day 5 post-infection. No clinical scores were observed in the mice vaccinated with 3 μg or 0.6 μg of COBRA rHA formulated with Infectimune^®^. All mice survived the H1N1 influenza virus infection except for one mouse in the adjuvanted 0.024 μg vaccinated group and one mouse in the adjuvanted 3 μg rHA vaccinated group. Clinical symptoms of illness were not observed in either of these mice at any timepoint, and both mice lost ~12–15% body weight during the infection (Figure 4A). Lungs were collected from three mice in each group on day 3 post-infection to quantify the live virus present in the respiratory tissues of the infected animals (Figure 4B). The pre-immune mock vaccinated animals had the most virus in their lungs on day 3, and the quantity of virus was significantly higher than the mice vaccinated with 3 μg or 0.6 μg of COBRA rHA formulated with Infectimune^®^ (*p* < 0.001). None of the mice vaccinated with 3 μg or 0.6 μg of adjuvanted COBRA rHA had any replicating virus in their lungs on day 3 post-infection, and mice vaccinated with 3 μg of unadjuvanted rHA had a similar amount of virus in their lungs to mice vaccinated with 0.12 μg of adjuvanted rHA (Figure 4B). In summary, mice vaccinated with 3 μg or 0.6 μg of CORA rHA adjuvanted with Infectimune^®^ had less virus in their lungs following infection, compared to animals immunized without adjuvant, which resulted in less weight loss, no clinical symptoms, and a faster recovery from disease.

### 4.3. COBRA rHA Vaccines Elicited Broader Antibody Responses than Wild Type rHA Formulated with Infectimune^®^ in Pre-Immune Ferrets

Pre-immune ferrets were vaccinated intramuscularly twice with a mixture of Y2 and NG2 COBRA rHA or a mixture of Mich/15 and Sing/16 wild type rHA formulated with Infectimune^®^ adjuvant. Two weeks after the second vaccination, sera samples collected from the ferrets were tested for HAI activity against panels of historical H1N1 and H3N2 vaccine strains (Figure 5). Every ferret vaccinated with COBRA Y2 and NG2 had antibodies with protective levels of HAI activity against all the H1N1 viruses from 2009 to 2019, with the highest antibody responses directed against the Mich/15 virus. The elicited HAI activity in these animals against Mich/15 and Bris/18 was significantly higher than the HAI activity against Vic/19 (*p* < 0.05) (Figure 5A). Ferrets vaccinated with the wild type rHAs also had mean protective serum HAI titers against the H1N1 influenza viruses isolated from 2009 to 2019, except for the Vic/19 strain. The highest antibody responses in these animals were directed against Mich/15, and these HAI titers were significantly higher than those elicited against Guang/19 (*p* < 0.05) and Vic/19 (*p* < 0.0001). These ferrets also had significantly higher HAI titers against the Cal/09 and Bris/18 isolates compared to Vic/19 (*p* < 0.001) (Figure 5B). The COBRA rHA vaccinated ferrets had mean protective HAI titers against all the H3N2 viruses in the panel isolated from 2013 to 2020 following the second vaccination, with the highest antibody responses directed toward SA/19. The HAI titers elicited in this group against SA/19 were statically higher than those elicited against Kan/17 (*p* < 0.001), HK/19 (*p* < 0.05), and Tas/20 (*p* < 0.05). The serum HAI activity in these animals was lowest against Kan/17, and these titers were statistically lower than HAI titers against HK/14 (*p* < 0.001), Sing/16 (*p* < 0.01), and Switz/17 (*p* < 0.001) (Figure 5D). Ferrets vaccinated with wild type rHA proteins formulated with Infectimune^®^ adjuvant had protective HAI titers against all the influenza viruses in the H3N2 panel, except for the Switz/13 and Kan/17 strains. The HAI titers elicited against these two viruses were statistically lower than HAI titers against every other strain in the H3N2 panel (*p* < 0.05–*p* < 0.001). The highest average antibody response in the wild type vaccinated animals was against the Switz/17 virus (Figure 5E). The mock vaccinated pre-immune ferrets did not have protective HAI titers against any of the H1N1 or H3N2 influenza viruses (Figure 5C,F).

The serum samples were pooled for each group of the vaccinated animals and were tested for their ability to neutralize viral infections in vitro (Figure 6). The COBRA and wild type HA vaccinated animals both had similar levels of neutralizing antibody titers (within ~2-fold) across the panel of H1N1 viruses, but the Y2 and NG2 rHA vaccinated animals had ~4-fold higher 50% neutralization titers against the Cal/09 and Guang/19 isolates (Figure 6A–E). The COBRA and wild type rHA vaccinated ferrets also had similar neutralizing antibody titers (within ~2-fold) across the panel of H3N2 viruses, but the ferrets vaccinated with wild type rHA had ~6-fold higher 50% neutralization titers against Sing/16 and ~4-fold higher titers against Tas/20 virus than the COBRA rHA vaccinated ferrets (Figure 6F–J).

The ferrets were infected intranasally with the A/Victoria/2570/2019 H1N1 influenza virus on day 116 of the study (Figure 7). None of the animals lost more than ~10% body weight during this infection and did not display any clinical signs of infection throughout the 14-day observation period. The ferrets administered Y2 and NG2 rHA vaccines only lost ~2% body weight on average, which peaked on day 5 post challenge. After day 5, this group gained weight for the duration of the infection, and some of the animals gained ~15% body weight during the observation period. The Mich/15 and Sing/16 rHA vaccinated animals lost ~5% body weight during the infection, which peaked on day 5 and returned near baseline by day 6 post-infection. The mock vaccinated pre-immune animals lost ~5% body weight on average by day 8 post-infection and did not start recovering body weight until day 11 (Figure 7A). Nasal wash samples were also collected from each animal on day 3 post-infection to determine the quantity of live virus present in the respiratory tract via plaque assay following infection. The Y2 and NG2 rHA vaccinated animals had no detectable virus in their nasal passages on day 3 post-infection. The Mich/15 + Sing/16 had ~7 × 10^2^ PFU/mL of live virus present in their nasal wash on day 3 post-infection, and the mock vaccinated animals had the most virus present in their nasal washes, averaging ~7.5 × 10^2^ PFU/mL (Figure 7B). Collectively, these data suggest that COBRA rHA vaccines adjuvanted with Infectimune^®^ elicit protective levels of HAI reactive antibodies against more historical and future drifted strains of H1N1 and H3N2 influenza viruses than wild type rHA vaccines while protecting animals from weight loss and preventing viral replication in the upper respiratory tract during an H1N1 influenza virus infection.

## 5. Discussion

Vaccine immunogenicity can be enhanced through the addition of adjuvants that work to either improve vaccine uptake or boost their ability to stimulate antigen-presenting cells (APCs) [37,38]. There are many different types of adjuvants, but only a limited number (Aluminum salts, MF-59, AS04, AS03, AS01, CpG ODN 1018, and lipid nanoparticles) are approved for human use [38]. In this study, Infectimune^®^, a cationic lipid nanoparticle adjuvant, was combined with soluble COBRA rHA antigens, administered to pre-immune mice, and effectively reduced the dose of vaccine necessary to elicit protective anti-influenza immune responses. Typically, lipid nanoparticles improve the recognition and endocytosis of vaccine antigens by APCs compared to antigens delivered as soluble proteins [31]. Increased uptake enhances antigen presentation by APCs and subsequently improves the production and quantity of high-affinity antibodies, thus allowing for dose sparing [38,39]. A dose-sparing effect was observed in this study as protective HAI titers against H1N1 influenza viruses were elicited in mice at a dose that was ~5–25× lower when rHA vaccines were formulated with Infectimune^®^ compared to unadjuvanted formulations of COBRA rHA. However, there was less of a dose-sparing effect observed against the H3N2 viruses where, in most cases, the COBRA rHA dose could only be reduced ~5× and still elicit protective HAI titers. These results may indicate an immunodominance effect between the H1 and H3 rHA antigens when administered simultaneously. Antigenic immunodominance is not uncommon in multivalent vaccinations and has been reported in other multivalent vaccines, such as those for dengue and human papillomavirus [40]. Multivalent influenza vaccination can induce competition between antigens, creating a hierarchy of strain-specific immunity that can be influenced by previous exposure to influenza antigens, relative protein abundance, or differential antigen processing [41,42]. Reduced antibody titers to H3N2, compared to other vaccine components, have also been observed in human responses to multivalent influenza vaccination [43]. However, similar antigenic immunodominance effects in trivalent influenza vaccination have been overcome in mice by increasing the dose of the sub-dominant antigen [44]. Whether or not the immunodominance observed during low-dose vaccinations is a result of the preferential recall of H1 memory cells from the first vaccination, relative protein abundance during vaccination, or differential antigen processing is unclear, but these immunodominance effects were less apparent when vaccinating pre-immune ferrets with the human vaccine dose of 15 μg. Overall, the Infectimune^®^ adjuvant enhanced the antibody responses elicited by the vaccines, as the unadjuvanted formulations were unable to elicit protective HAI titers against most of the viruses in the H1N1 and H3N2 influenza virus panels. In general, pre-immune mice vaccinated with 3 μg of unadjuvanted rHA had similar HAI and ELISA titers to animals immunized with an ~25-fold lower dose (0.12 μg) of rHA formulated with Infectimune^®^.

In addition to increasing antibody titers, adjuvants can also help drive the production of various antibody isotypes, leading to differential immune responses [45]. Typically, oil-in-water emulsion or aluminum-based adjuvants will preferentially generate IgG1 antibody responses associated with Th2-type humoral immunity [38]. The mice in this study vaccinated with the Infectimune^®^ adjuvant had similar levels of IgG1 and IgG2a-based antibody responses, suggesting that they have a balanced cellular Th1 and humoral Th2 immune response [46]. Cellular-based immunity is an important immunological component that can increase the protective efficacy of many vaccine candidates through a multitude of different effector functions. CD4+ T cells, in particular, have a broad specificity that can target many different peptide epitopes present on viral proteins and also promote the expansion of CD8+ T cells through the production of IFN-γ and IL-2 [47,48]. Mice vaccinated with Flublok^®^ and COBRA rHA formulated with Infectimune^®^ were previously shown to elicit higher levels of IFN-γ and IL-2 secreting epitope-specific CD4+ T cells when compared to emulsion-based adjuvants [30,49,50]. Thus, as observed in this study, a combination of cellular Th1 and humoral Th2 immunity was likely responsible for helping to prevent the virus from propagating in the host’s respiratory tissue and protect mice from H1N1 influenza virus challenge, even when they were vaccinated with reduced antigen doses. However, all the animals used in this study were female, which tend to have a Th2 bias in response to influenza vaccination and different results may be observed in male animals [51,52].

Immune responses elicited by vaccination can also be influenced by the antigens included in the vaccine and an individual’s exposure history to past influenza viruses. Most people encounter influenza either through vaccination or infection by the time they are three years old [8,53,54]. In this study, a human-like immune state was modeled by exposing mice and ferrets to historical strains of H1N1 and H3N2 influenza viruses prior to vaccination. The pre-immune ferrets that were vaccinated with either wild type or COBRA rHA had protective HAI antibody titers against several historical H1N1 and H3N2 influenza virus vaccine strains. However, the ferrets vaccinated with the COBRA rHA antigens elicited protective antibodies against more of these viral strains than the animals vaccinated with the wild type rHA antigens. This is likely a result of the recall of memory immune cells that are specific to common epitopes shared between the COBRA rHA, the viruses used to establish pre-immunity, and the HA antigens on the surface of the viruses in the HAI panel that are not present on the wild type H1 and H3 rHA vaccine antigens. Since COBRA HA antigens are designed from a diverse pool of co-circulating HA sequences, they likely include a variety of immunologic epitopes that are present on multiple antigenically drifted strains. Previous iterations of COBRA HA molecules elicited polyclonal pools of broadly reactive antibodies that are specific to both the HA head and stem regions, and current studies characterizing the antibodies elicited by the Y2 and NG2 antigens are in progress [55,56]. Nevertheless, the mice and ferrets in this study were only exposed to one strain of H1N1 and H3N2 influenza and do not have the diverse memory cell repertoires that a human would have after a lifetime of infections and vaccinations. Therefore, it is possible that both the COBRA and wild type rHA vaccines would perform differently in a population that has a more extensive exposure history to influenza than what was modeled in this study.

The Y2 rHA vaccinated ferrets in this study elicited antibodies with higher HAI activity against the Vic/19 strain than the Mich/15 rHA vaccinated ferrets. The Y2 rHA vaccinated ferrets also experienced little to no weight loss and had no detectable virus present in their nasal passages following infection with the Vic/19 virus. In comparison, the Mich/15 rHA vaccinated animals had an ~2-fold higher amount of virus in their nasal wash samples post-infection than the ferrets vaccinated with COBRA rHA, suggesting that the COBRA rHA vaccine was more protective than the wild type vaccine against an antigenically drifted H1N1 virus. The amino acid sequences of the Y2 and Mich/15 HA only differ in 4 locations, and in 3 of those positions, the Mich/15 and Vic/19 HA proteins have identical amino acids. At site 312, both Y2 and Vic/19 HA proteins have a valine, while Mich/15 has an isoleucine. Site 312 is in the stem region of the HA molecule near the interface of the HA head and stem regions. Most HA stem-directed antibodies do not have HAI activity; thus, it is unlikely that these sites are playing a direct role in the observed differences in HAI reactivity, but they could be contributing to the observed differences in neutralization titer [57,58]. However, a combination of amino acid substitutions in sites 179 and 181, which are located in antigenic site Sa, may be driving the observed differences in HAI reactivity [59]. In this location, both the Mich/15 and Vic/19 HA proteins have a *N*-linked glycosylation motif, N-X-S/T, where X represents any amino acid other than proline, but they differ at site 181, where the Mich/15 HA has a serine and Vic/19 HA has a threonine. The Y2 COBRA HA lacks a glycosylation motif at this location. Therefore, even though the Vic/19 HA possesses a glycosylation motif in antigenic site Sa, it is possible that there is no glycan present, and it exposes a similar epitope as the Y2 HA in this location.

In the mouse model, the NG2 rHA vaccine-elicited protective HAI titers against most of the H3N2 viruses in the panel. The lowest antibody responses in this model were directed against the Kan/17 and HK/19 strains. However, in the ferret model, the NG2 rHA elicited protective HAI titers against all the H3N2 viruses. While the NG2 COBRA rHA did not elicit HAI titers that were greater in magnitude than the Sing/16 rHA vaccinated ferrets, all the ferrets vaccinated with NG2, except for one animal against the HK/19 strain, had protective HAI titers against every virus. Additionally, the Sing/16 rHA vaccinated ferrets lacked protective HAI titers against the Switz/13 and Kan/17 strains, and the NG2 COBRA rHA elicited protective HAI titers against these strains that were ~2–4 times higher than the Sing/16 rHA. The neutralizing antibody titers against Kan/17 were higher in the ferrets vaccinated with wild type rHA compared to the COBRA rHA vaccinated animals, but since the samples analyzed were pooled it is possible that the one ferret that had a high HAI titer against the Kan/17 virus is raising the neutralization titer of the pooled sample. Both of these H3N2 viruses, Switz/13 and Kan/17, are members of clade 3c.3a, and the Sing/16 virus belongs to clade 3c.2a1 [3]. Typically, viruses in these two clades are antigenically distinct, as strains in clade 3c.2a have a glycosylation at site 160 that may mask epitopes that are exposed on 3c.3a isolates [60,61]. However, both the NG2 and Sing/16 HA proteins have glycosylation motifs at residue 160, which consists of the same amino acids. The Sing/16 and NG2 HA proteins differ by 5 amino acids, and only one of those, residue 187, is in the HA head region. In this location, the Sing/16 HA has a lysine (a charged, aliphatic amino acid), and NG2 HA shares asparagine (a polar, aliphatic amino acid) with Switz/13 and Kan/17. Residue 187 is in antigenic site B, and historically, single amino acid substitutions in this region have been associated with major antigenic changes [60]. Therefore, it is likely that the lysine in the Sing/16 HA at site 187 is responsible for the observed differences in HAI reactivity. Future experiments will focus on making substitutions to replace the 187K in the Sing/16 HA and remove the glycosylation at site 179 in the Mich/15 HA to determine if these can enhance the immunologic breadth of these antigens.

## 6. Conclusions

Altogether, the combination of COBRA rHA and Infectimune^®^ investigated in this study was effective at eliciting protective levels of antibodies across a panel of H1N1 and H3N2 influenza vaccine strains isolated over the last decade in both pre-immune mice and ferrets. The addition of the adjuvant enhanced the antibody titers, compared to unadjuvanted groups and allowed for a dose-sparing effect. Future studies will focus on optimizing the antigen dose to achieve a more balanced immune response among the vaccine components, while this study highlights the potential benefits of combining broadly reactive antigens with potent immunostimulatory molecules.

## Figures and Tables

**Figure 1 vaccines-12-01364-f001:**
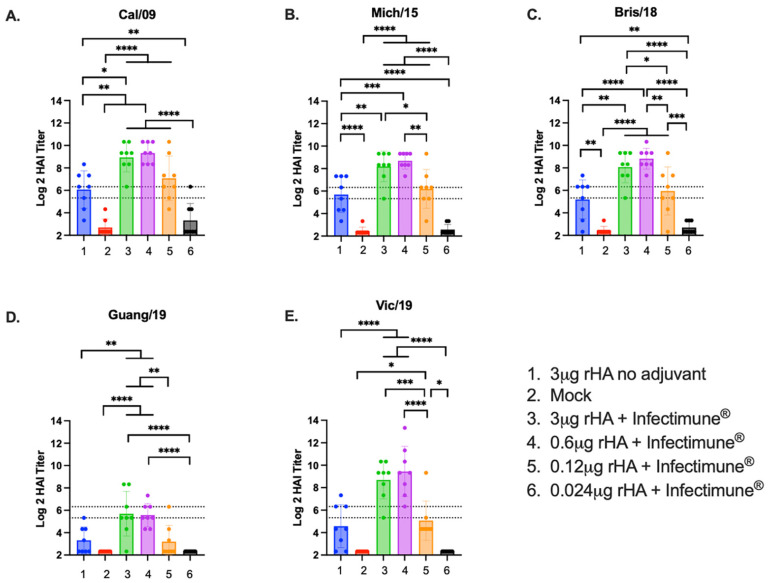
Mouse H1N1 HAI panel. HAI assays were performed using serum collected from each mouse on day 72 of the study against a panel of H1N1 viruses. The H1N1 panel consisted of 5 viruses isolated from 2009 to 2019 and are listed at the top of each graph (**A**–**E**). The Log2 HAI titer is reported on the y-axis. The lower dotted line on the y-axis represents an HAI titer of 1:40, and the upper dotted line represents a titer of 1:80. The vaccine groups are listed on the x-axis of each figure (1–6). (1) 3 μg rHA no adjuvant (blue). (2) Pre-immune mock vaccinated (red). (3) 3 μg rHA + Infectimune^®^ (green). (4) 0.6 μg rHA + Infectimune^®^ (purple). (5) 0.12 μg rHA + Infectimune^®^ (orange). (6) 0.024 μg rHA + Infectimune^®^ (black). Statistical values were determined via one-way ANOVA (* = *p* < 0.05, ** = *p* < 0.01, *** = *p* < 0.001, **** = *p* < 0.0001).

**Figure 2 vaccines-12-01364-f002:**
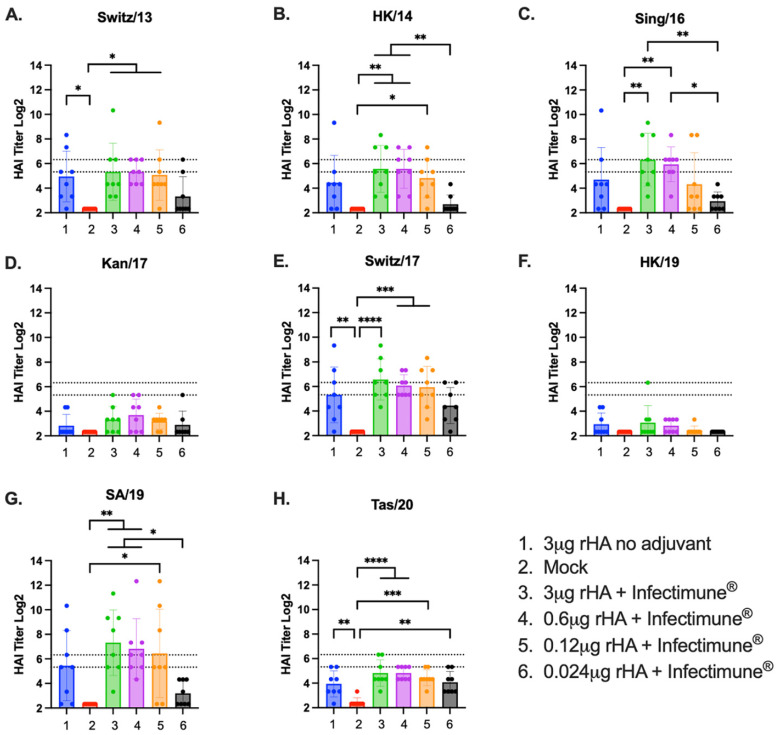
Mouse H3N2 HAI panel. HAI assays were performed using serum collected from each mouse on day 72 of the study against a panel of H3N2 viruses. The H3N2 panel consisted of 8 viruses isolated from 2013 to 2020 and are listed at the top of each graph (**A**–**H**). The Log2 HAI titer is reported on the y-axis. The lower dotted line on the y-axis represents an HAI titer of 1:40, and the upper dotted line represents a titer of 1:80. The vaccine groups are listed on the x-axis of each figure (1–6). (1) 3 μg rHA no adjuvant (blue). (2) Pre-immune mock vaccinated (red). (3) 3 μg rHA + Infectimune^®^ (green). (4) 0.6 μg rHA + Infectimune^®^ (purple). (5) 0.12 μg rHA + Infectimune^®^ (orange). (6) 0.024 μg rHA + Infectimune^®^ (black). Statistical values were determined via one-way ANOVA (* = *p* < 0.05, ** = *p* < 0.01, *** = *p* < 0.001, **** = *p* < 0.0001).

**Figure 3 vaccines-12-01364-f003:**
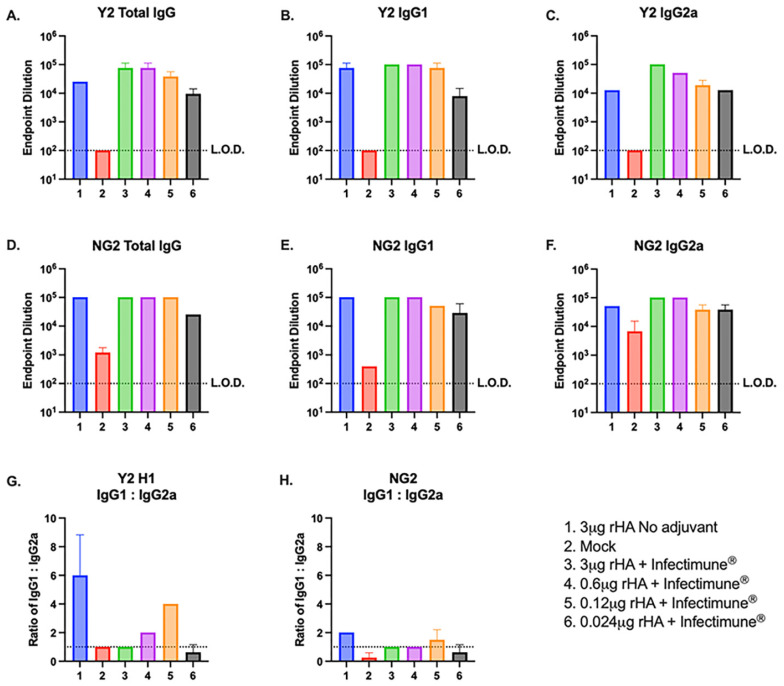
Mouse total IgG and isotype ELISA tiers. ELISA assays detecting the presence of total IgG or different antibody isotypes were performed using serum collected from each mouse on day 72 of the study that was pooled for each group. Plates were coated with either Y2 (**A**–**C**) H1 or NG2 (**D**–**F**) H3 rHA and probed with different secondary antibodies specific for either total IgG (**A**,**D**), IgG1 (**B**,**E**) or IgG2a (**C**,**F**). Endpoint dilution titers are plotted on the y-axis, and the vaccine groups are listed on the x-axis of each figure (1–6). (1) 3 μg rHA no adjuvant (blue). (2) Pre-immune mock vaccinated (red). (3) 3 μg rHA + Infectimune^®^ (green). (4) 0.6 μg rHA + Infectimune^®^ (purple). (5) 0.12 μg rHA + Infectimune^®^ (orange). (6) 0.024 μg rHA + Infectimune^®^ (black). The ratio of the IgG1:IgG2a endpoint dilution titers were determined for the H1-specific anti-Y2 antibodies (**G**) and H3 specific anti-NG2 antibodies (**H**) for each group of vaccinated mice.

**Figure 4 vaccines-12-01364-f004:**
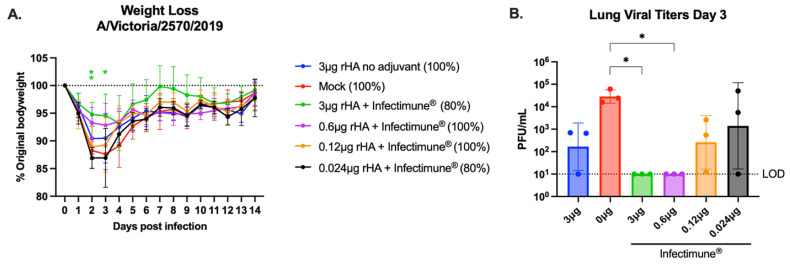
Mouse challenge weight loss and viral plaque titers. Mice were infected with A/Victoria/2570/2019 H1N1 virus on day 86 of the study. The mice were monitored for weight loss and survival (% listed next to each vaccine group) (**A**). On day 3 following infection, lungs were harvested from 3 mice in each group, which were assessed for viral titers via plaque assay (**B**). The different groups are represented in each figure by different colors: 3 μg rHA (blue), pre-immune mock vaccinated (red). 3 μg rHA + Infectimune^®^ (green). 0.6 μg rHA + Infectimune^®^ (purple). (5) 0.12 μg rHA + Infectimune^®^ (orange). (6) 0.024 μg rHA + Infectimune^®^ (black). Statistical values were determined via one-way ANOVA (* = *p* < 0.05, ** = *p* < 0.01).

**Figure 5 vaccines-12-01364-f005:**
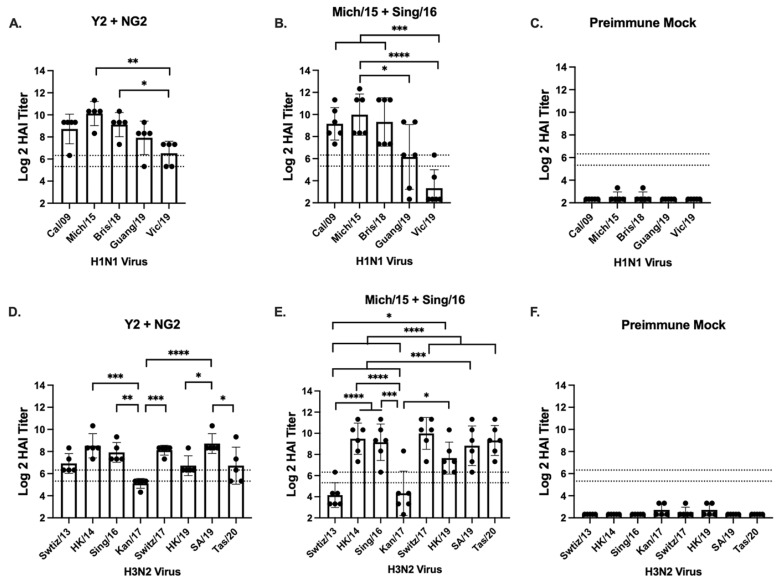
Ferret H1N1 and H3N2 HAI panel. HAI assays were performed using serum collected from each ferret on day 102 of the study against a panel of H1N1 (**A**–**C**) and H3N2 (**D**–**F**) viruses. The H1N1 panel consisted of 5 viruses isolated from 2009 to 2019, and the H3N2 panel consisted of 8 viruses isolated from 2013 to 2020. The vaccine groups are listed at the top of each figure. The Log2 HAI titer is reported on the y-axis. The lower dotted line on the y-axis represents an HAI titer of 1:40, and the upper dotted line represents a titer of 1:80. The viruses in each panel are listed on the x-axis of each figure. Statistical values were determined via one-way ANOVA (* = *p* < 0.05, ** = *p* < 0.01, *** = *p* < 0.001, **** = *p* < 0.0001).

**Figure 6 vaccines-12-01364-f006:**
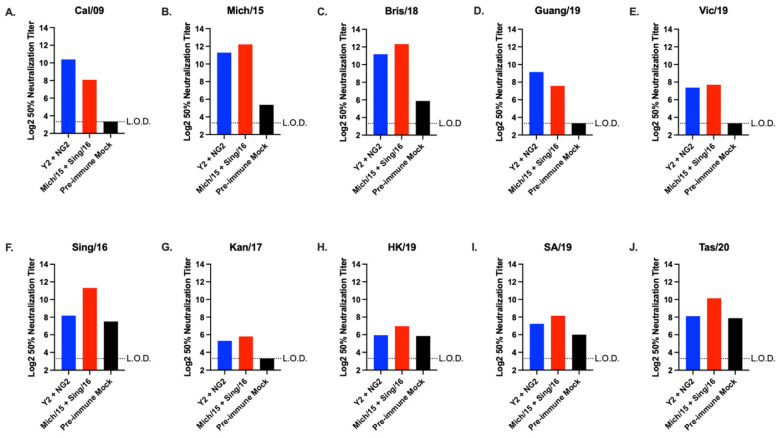
Ferret H1N1 and H3N2 Microneutralization panel. Microneutralization assays were performed using pooled serum collected from each group of ferrets on day 102 of the study against a panel of H1N1 and H3N2 viruses. The H1N1 panel consisted of 5 viruses isolated from 2009 to 2019 (**A**–**E**), and the H3N2 panel consisted of 5 viruses isolated from 2016 to 2020 (**F**–**J**). The Log2 50% neutralization titers are listed on the y-axis of each figure. The vaccine groups are listed on the x-axis of each figure. The virus used in each assay is listed at the top of each figure. Data from each group are represented by different colors: Y2 + NG2 (blue), Mich/15 + Sing/16 (red), and pre-immune mock vaccinated (black).

**Figure 7 vaccines-12-01364-f007:**
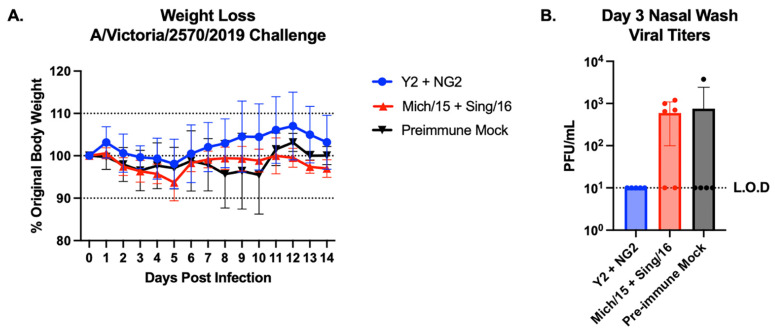
Ferret challenge weight loss and viral plaque titers. The vaccinated pre-immune ferrets were infected with A/Victoria/2570/2019 H1N1 virus on day 116 of the study and were tracked for weight loss for 14 days (**A**). Weight loss values were reported as percent original body weight based on the weight of each animal on day 116 and are listed on the y-axis. On day 3 following infection, nasal wash samples were collected from the ferrets in each group, which were assessed for viral titers via viral plaque assay (**B**). Viral titers of the nasal wash samples are reported as PFU/mL on the y-axis. Data from each group are represented by different colors: Y2 + NG2 (blue), Mich/15 + Sing/16 (red), and pre-immune mock vaccinated (black).

## Data Availability

The original data presented in the study are made publicly available by the National Institutes of Health ImmPort online database.

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
