# Peer review of "Computationally Optimized Hemagglutinin Proteins Adjuvanted with Infectimune® Generate Broadly Protective Antibody Responses in Mice and Ferrets"

_vaccines, 2024, doi:10.3390/vaccines12121364_

Round 1

Reviewer 1 Report

Comments and Suggestions for Authors

Dear authors:

The manuscript entitled "Computationally optimized hemagglutinin proteins adjuvanted with infectimune  generate broadly protective antibody responses in mice and ferrets" by Allen et al. is well written and presents information about two COBRA generated vaccines, Y2 (H1H1) and NG2 (H3N2), in combination with the adjuvant Infectimune which provides important contributions to vaccine research.

Major points:

The study employs only female mice and ferrets. With respect to the immune system there are significant differences between males and females. Some of these differences could be reflected in the Th1 (IgG2a) versus Th2 (IgG1) results present in this paper. Usually, females tend to be more skewed towards Th2.

With respect to vaccination of the mice with Y2 + NG2 this vaccine did not increase the titer much above 1:40 for any of the H3N2 challenge viruses. The neutralizing antibody titers were especially low for the Kan/17 and HK/19 viruses. An attempt to explain these results was made in the discussion.

The weak performance of the NG2 component was also displayed by the results presented for the Ferret experiments. In this case neutralization titers were higher for ferrets immunized with the Mich/15 + Sing/16 vaccine than for the Y2 + NG2 vaccine after challenge with H3N2 viruses.  Again, these results were duplicated with the microneutralization assay.

Interestingly, for the experiments performed with ferrets, the Y2 + NG2 vaccine resulted in a marked reduction in viral plaques than immunization with the Mich/15 + Sing/16 combination. This result could have been further elaborated on in the discussion.

In line 626 the "H3N3" in think should be "H3N2".

In the abstract the font size for the Conclusions and Key words should be increased.

Author Response

Major points:

  1. The study employs only female mice and ferrets. With respect to the immune system there are significant differences between males and females. Some of these differences could be reflected in the Th1 (IgG2a) versus Th2 (IgG1) results present in this paper. Usually, females tend to be more skewed towards Th2.

  • Thank you for your comment. A statement has been added to the discussion to include the caveat that only female animals were used in this study and different results may have been seen in male animals.  See lines 964-965.

  1. With respect to vaccination of the mice with Y2 + NG2 this vaccine did not increase the titer much above 1:40 for any of the H3N2 challenge viruses. The neutralizing antibody titers were especially low for the Kan/17 and HK/19 viruses. An attempt to explain these results was made in the discussion.

–    Thank you for your comment.  The HAI titers were low for the mice, but in the ferret model the titers reached protective levels for all animals except 1 against the Kan/17 virus and in all animals against HK/19.  Text can has been added to the discussion to emphasize this.  See lines 1000 – 1007.

  1. The weak performance of the NG2 component was also displayed by the results presented for the Ferret experiments. In this case neutralization titers were higher for ferrets immunized with the Mich/15 + Sing/16 vaccine than for the Y2 + NG2 vaccine after challenge with H3N2 viruses.  Again, these results were duplicated with the microneutralization assay.

 –      Thank you for your comment.  The magnitude of the HAI titers were higher in the wild type vaccinated ferrets against most of the strains, but they were not at protective levels against all of the H3N2 viruses, like what was observed in the NG2 vaccinated animals.  The average HAI titers of the wild type vaccinated animals were well below protective levels against the Switz/13 and Kan/17 viruses.  Since samples for the neutralization assay were pooled, it is possible that the one ferret that had a high HAI titer against the Kan/17 virus is pulling up the neutralization titers of the sample against this virus. Text has been added to the discussion to emphasize these points.  See lines 1000 – 1010.

  1. Interestingly, for the experiments performed with ferrets, the Y2 + NG2 vaccine resulted in a marked reduction in viral plaques than immunization with the Mich/15 + Sing/16 combination. This result could have been further elaborated on in the discussion. – Thank you for pointing this out. Text has been added to the discussion to emphasize the lack of viral replication in the nasal passages in the COBRA vaccinated animals compared to the wild-type vaccinated animals.  See lines 982 – 987.

  1. In line 626 the "H3N3" in think should be "H3N2".

– Thank you for pointing this out. The text has been updated.  See line 1039.

  1. In the abstract the font size for the Conclusions and Key words should be increased.

– Thank you for pointing this out. The text has been updated.  See lines 33-40.

Please see the attached manuscript with tracked changes.

Reviewer 2 Report

Comments and Suggestions for Authors

Review on: “Computationally optimized hemagglutinin proteins adjuvanted with Infectimune® generate broadly protective antibody responses in mice and ferrets 

The objective of the study was to evaluate the immunogenic and protective properties of the computationally optimized recombinant hemagglutinin (rHA) supplemented with adjuvant Infectimune®. The study was conducted on mice and ferrets. It was shown that computationally optimized rHA vaccines adjuvanted with Infectimune® elicit protective levels of HAI reactive antibodies against more drifted strains of H1N1 and H3N2 influenza viruses than wild type rHA vaccines. After challenge with A/Victoria/2570/2019 H1N1 virus animals were protected from weight loss and had decreased viral replication in the upper respiratory tract. The article is well-designed. The methods are written in great detail, giving readers the opportunity to repeat the experiments. A large number of figures are provided with detailed descriptions.

Note

1)     Section “Vaccination and viral infection of mice” line 143-152

It would be more convenient for readers if the characteristics of the reagents and Infectimune® adjuvant were moved above, to the “Materials” section, so that the description of the groups would be more compact.

2)     Figure 1. A.  The high HAI titer for group 6 must be discussed (If it is not mistake)

3)     Figure 4. A.  It is unclear what the curve (Зµg rHA + Infectimune" is compared to with **=p <0.01

Author Response

Major Points

  1. Section “Vaccination and viral infection of mice” line 143-152.  It would be more convenient for readers if the characteristics of the reagents and Infectimune® adjuvant were moved above, to the “Materials” section, so that the description of the groups would be more compact.

 – Thank you for your comment.  The text has been modified to accommodate this request.  See lines 117 - 120

  1. Figure 1. A.  The high HAI titer for group 6 must be discussed (If it is not mistake).

– Thank you for pointing this out.  We went back and checked the raw data, and an error was made during data entry.  The text has been updated on lines 571 - 572, an updated figure has been inserted into the text, and new raw data files have been provided to the journal.

  1. Figure 4. A.  It is unclear what the curve (Зµg rHA + Infectimune" is compared to with **=p <0.01.

– Thank you for bringing this to our attention. The text has been updated to clarify this point.  See lines 791 – 792.

Please see the attached revised manuscript with tracked changes.